# Dietary Recommendations for Bariatric Patients to Prevent Kidney Stone Formation

**DOI:** 10.3390/nu12051442

**Published:** 2020-05-16

**Authors:** Milene S. Ormanji, Fernanda G. Rodrigues, Ita P. Heilberg

**Affiliations:** 1Nephrology Division, Universidade Federal de São Paulo, São Paulo 04023-062, Brazil; milene.ormanji@gmail.com (M.S.O.); fernanda.gr91@gmail.com (F.G.R.); 2Department of Nutrition, Universidade Federal de São Paulo, São Paulo 04023-062, Brazil

**Keywords:** nephrolithiasis, bariatric surgery, hyperoxaluria, kidney stones, diet

## Abstract

Bariatric surgery (BS) is one of the most common and efficient surgical procedures for sustained weight loss but is associated with long-term complications such as nutritional deficiencies, biliary lithiasis, disturbances in bone and mineral metabolism and an increased risk of nephrolithiasis, attributed to urinary metabolic changes resultant from low urinary volume, hypocitraturia and hyperoxaluria. The underlying mechanisms responsible for hyperoxaluria, the most common among all metabolic disturbances, may comprise increased intestinal oxalate absorption consequent to decreased calcium intake or increased dietary oxalate, changes in the gut microbiota, fat malabsorption and altered intestinal oxalate transport. In the current review, the authors present a mechanistic overview of changes found after BS and propose dietary recommendations to prevent the risk of urinary stone formation, focusing on the role of dietary oxalate, calcium, citrate, potassium, protein, fat, sodium, probiotics, vitamins D, C, B6 and the consumption of fluids.

## 1. Introduction

Obesity is one of the most important worldwide public health challenges predisposing to severe comorbidities such as diabetes mellitus, cardiovascular disease, cancer, sleep apnea and hypertension [1]. Considering the difficulties regarding diet therapy as a long-term control of morbid obesity, bariatric surgery (BS) translated into an efficient method for sustained weight loss [2].

BS procedures comprise restrictive techniques like gastric banding and sleeve gastrectomy, malabsorptive techniques such as biliopancreatic diversion and duodenal switch, or a combination of both as in a Roux-en-Y gastric bypass (RYGB), one of the most common surgical procedures performed over the last years. Although BS is considered an efficacious technique with benefits concerning the treatment of comorbidities of the morbidly obese patients, it may bring long-term complications such as nutritional deficiencies, biliary lithiasis, disturbances in bone and mineral metabolism and an increased risk of nephrolithiasis [3,4]. Such a risk is estimated to be around 7.6% in bariatric patients until 5 years after surgery, which represents almost a two-fold increase in risk if compared with morbidly obese patients [5,6], showing the greatest risk in malabsorptive procedures, intermediate risk in standard RYGB, and the least risk in restrictive techniques [7].

Nephrolithiasis arises from urinary metabolic changes in these patients, such as low urinary volume, hypocitraturia and hyperoxaluria. As a rule, hyperoxaluria may be due to the rare monogenic disorders of primary hyperoxaluria, characterized by increased production of endogenous oxalate within the liver consequent to the various mutations as opposed to secondary hyperoxaluria, ascribed to excessive oxalate intake or increased intestinal absorption (enteric). Secondary hyperoxaluria is certainly the case in patients that have undergone BS and represents the most frequent metabolic disturbance detected among them, with prevalence rates ranging from 29% to around 67% at 3 months and 2 years after BS [8,9,10,11]. A recent meta-analysis has demonstrated a 36.4% increase in urinary oxalate levels after BS considering the 24-h urine profile from 277 patients belonging to six prospective studies after almost 1 year of RYGB [12].

The underlying mechanisms for an increasing urinary oxalate in post-BS patients have not been completely elucidated but may be accounted for dietary factors, intestinal fat malabsorption, alterations in gut microbiota, and/or changes in the intestinal oxalate transport [13], as hypothesized in Figure 1 below:

A diet rich in oxalate and/or poor in calcium decreases the generation of unabsorbable calcium oxalate (CaOx) complexes ultimately leading to a higher amount of free oxalate in the intestinal lumen. In a previous study conducted by our group [14], an exaggerated oxaluric response was observed following an oral oxalate load in 61 post-BS patients (58 RYGB and 3 biliopancreatic diversion with duodenal switch) compared to morbidly obese patients and also to their own urinary oxalate levels 6 months before the procedure, suggesting an increased absorption of dietary oxalate as a predisposing mechanism for enteric hyperoxaluria.

The occurrence of hyperoxaluria following BS can be also associated with increased fecal fat malabsorption, probably due to the higher amount of unabsorbed bile and fatty acids which saponify intestinal calcium, limiting the amount of luminal free-calcium binding with oxalate [15,16,17,18].

Alterations in intestinal microbiota induced by the intestinal bypass such as a reportedly reduced colonization by *Oxalobacter formigenes*, commensal Gram-negative anaerobic bacteria with naturally-occurring oxalate metabolizing properties [19,20,21], could also contribute to increased oxaluria after BS. However, the effects of *O. formigenes* intestinal colonization upon urinary oxalate remain controversial [22] both in experimental and clinical settings. In an experimental model of hyperoxaluric RYGB in rats, animals colonized with *O. formigenes* were able to reduce 74% of their urinary oxalate [23]. However, although a clinical study has shown that the colonization with *O. formigenes* has been associated with a substantial reduction in the risk of recurrent stone formation among non-obese patients, urinary oxalate did not differ with the presence or absence of *O. formigenes* colonization [24]. Among morbidly obese patients, although 84% were not colonized by these bacteria, their urinary oxalate did not differ from the ones who were colonized [25]. More recently, a collaborative relationship between *O. formigenes* and other bacterial species in intestinal oxalate homeostasis in individuals with or without urinary stone disease has been suggested [26]. Of note, in a small series of bariatric patients, Froeder et al. [14] did not observe less colonization by *O. formigenes* in fecal samples analyzed by PCR. 

Unabsorbed intestinal bile and fatty acids could cause modifications in intestinal tight junctions leading to increased intestinal permeability and consequent increased passive oxalate transport from the intestine into the bloodstream [27]. In an experimental study, Hatch et al. [28] observed that the RYGB procedure altered the permeability of the colon to oxalate, promoting higher intestinal oxalate paracellular absorption.

In experimental studies, Freel et al. [29,30] demonstrated that knockout (KO) mice for the intestinal oxalate exchanger responsible for active Ox secretion, Slc26a6 (PAT1), exhibited higher urinary oxalate excretion whereas the KO model for Slc26a3 (DRA), the exchanger which mediates Ox reabsorption, presented lower urinary oxalate when compared to wild-type animals. Nevertheless, in a model of mini-gastric bypass surgery in rats fed with oxalate and fat previously developed in our laboratory, no changes in the intestinal expression of Slc26a3 and Slc26a6 had been demonstrated [31].

Another long-term complication of BS comprises disturbances in bone and mineral metabolism such as decreased bone mass and increased risk of fractures because of mechanical loading decrease, loss of lean body mass loss, hypovitaminosis D, malabsorption of calcium, vitamin D and other nutritional deficiencies as well as hormonal changes following the procedure [32,33,34]. Several investigators have reported an increase in markers of bone turnover [4,35,36] and reduced bone mineral density [37]. In a recent study, Melo et al. [4] demonstrated an increase in both bone formation and resorption markers among BS patients up to more than 7 years after the surgical procedure, suggesting that an increased bone turnover persists even at a very long-term period of follow-up.

## 2. Dietary Recommendations 

### 2.1. Oxalate

Although a low-oxalate diet is recommended to prevent hyperoxaluria and stone formation after BS, the lack of information about oxalate content in foods can be an obstacle while trying to restrict oxalate from the diet. So far, the most complete and trustworthy database of foods analyzed for oxalate content has been provided by Harvard School of Public Health [38]. A list of foods rich in oxalate may include spinach, rhubarb, beets, starfruit, okra, some nuts, dark chocolate, legumes (ex. beans, soy), dark tea, parsley and some berries, with the first two being the richest (Table 1) [38,39,40,41,42]. In epidemiological studies, Taylor et al. [43] showed that raw and cooked spinach represented around 40% of the oxalate intake in their cohorts. According to the American Dietetic Association (ADA) the oxalate intake recommendation is around 40–50 mg/day [44]. However, the oxalate intake from a typical North-American diet remains between 150 to 200 mg/day [40,43] irrespective of the history of stone formation. In non-Western diets, oxalic acid consumption was reported to differ a lot according to the season, socioeconomic level, urban versus rural areas, eventually reaching amounts as high as 2000 mg/day [45]. In our outpatient unit, we also did not observe differences between stone-formers (SF) and non-SF with respect to oxalate intake with a mean consumption of around 100 mg/day for both [46,47]. Absorption of dietary oxalate is highly variable between individuals as well as its contribution to the urinary oxalate excreted [40] and the food oxalate content can vary due to different methods of cooking and agricultural conditions, such as soil and water nutrients [48]. Furthermore, bioavailability of oxalate in foods is highly variable [49] so that the increment in oxaluria may be greater even after consumption of foods containing a smaller oxalate content but in a more soluble form [50]. Oxalate absorption from almonds has been reported to be six-fold higher than from black beans [51]. Moreover, intestinal oxalate absorption is known to be critically dependent on the amount of daily calcium intake [47,52,53], increasing from around 3% with high calcium intakes to as much as 17% under conditions of a very low intake [52], which can be the case among bariatric patients. Dietary habits of patients after BS may or may not change, thus exposing them to such issues. Froeder et al. [14] have shown a two-fold increase in oxaluria by BS patients after the consumption of a spinach juice containing 375 mg of oxalate in comparison to morbidly obese subjects, reinforcing the hypothesis of enteric hyperoxaluria.

In summary, although there is no consensus on the grade of dietary oxalate restriction by bariatric patients, there is enough data to support the idea that they should at least limit the foods with the highest oxalate content, as shown below in Table 1, while maintaining a proper calcium intake to counteract the absorption of oxalate

### 2.2. Calcium and Vitamin D

Calcium absorption predominantly occurs in the duodenum and proximal jejunum and is dependent on vitamin D levels. Due to fat malabsorption, all fat-soluble vitamins (A, D, E and K) are at risk of deficiency among bariatric patients. Some investigators pointed out a deficient calcium intake and vitamin D deficiency after RYGB [60]. Schafer et al. [61] demonstrated that even patients with acceptable levels of vitamin D (≥ 30 ng/mL) and maintained under an adequate calcium intake (> 1200 mg/day) had a marked decrease in intestinal calcium absorption from 33% preoperatively to 7% after 6 months of RYGB. The common use of proton-pump inhibitors by bariatric patients may also affect calcium absorption contributing to the exacerbation of such deficiency [62]. Clinical studies have also identified hypovitaminosis D in morbidly obese patients prior to RYGB [63,64]. According to the Clinical Practice Guidelines for the Perioperative Nutritional, Metabolic, and Nonsurgical Support of the Bariatric Surgery Patient [65], calcium supplementation should be at least 1200–1500 mg/day consisting of the usual consumption of calcium-rich foods like dairy products, seafood, almonds, green vegetables and other food items fortified with calcium. However, the consumption of milk and other dairy products have been associated with “Dumping syndrome” in some patients. The latter is due to the rapid emptying of food into the small intestine triggering rapid fluid shifts into the intestinal lumen and release of gastrointestinal hormones, causing gastrointestinal and vasomotor symptoms such as bloating, nausea, diarrhea, dizziness and sweating, among others [66]. The natural sugar in dairy products (lactose) might worsen such symptoms which may appear soon after eating or later. Besides diet and especially for such patients, the use of adequate calcium supplements in the form of citrate salts is mandatory as gastric acid secretion might be reduced after BS averting the absorption of calcium carbonate. Furthermore, another advantage of calcium citrate supplementation is the reduction of urinary phosphate that, in association with the inhibitory effects of citrate, might protect against stone formation [67]. In fact, a randomized, double-blind crossover study of RYGB patients, confirmed a better bioavailability of calcium citrate than calcium carbonate [68]. The calcium bioavailability of a formulation of effervescent potassium calcium citrate after RYGB has been shown to be useful as well [69]. Notably, hypercalciuria is not a frequent finding in post-BS patients [70]. Nonetheless, taking the calcium supplements is preferable with meals, hence helping to prevent increases in urinary CaOx supersaturation. With respect to Vitamin D, a specific study in post-bariatric pregnant women reported that women after the first year of RYGB may present increased vitamin D demands compared to pregnant women who did not undergo surgery [71]. According to the American guideline for BS, such patients should have nutritional surveillance and laboratory screening for deficiency every trimester [65]. The minimal daily vitamin D supplementation for BS patients is at least 3000 international units (IU) until blood levels are greater than 30 ng/mL and in cases of severe vitamin D malabsorption, 50,000 UI for 1 to 3 times weekly to daily [65,72]. The European Guideline also suggests a supplementation of 1200–1500 mg of elemental calcium (in diet and/or as citrate supplements in divided doses), and at least 3000 IU of vitamin D per day for post-BS patients [73]. Regardless of the difficulties concerning the adequate intestinal absorption after BS, calcium supplementation and vitamin D repletion are essentially required to prevent both nephrolithiasis and bone disease.

In summary, the recommended amount of calcium intake after BS should be at least 1200–1500 mg/day, provided by diet or supplements and at least 3000 IU of Vitamin D per day adjusting to maintain adequate serum levels.

### 2.3. Vitamin B6

Considering endogenous metabolism, vitamin B6 (pyridoxine), in the form of pyridoxal phosphate, is a required cofactor of the enzyme alanine-glyoxylate aminotransferase (AGT) for the transamination of glyoxylate to glycine. When vitamin B6 status is inadequate for enzyme activity, a higher amount of glyoxylate is converted to oxalate by the lactate dehydrogenase [74]. The current recommended dietary allowance (RDA) for vitamin B6 is around 1.3 mg/day for healthy individuals [75] and the richest sources of vitamin B6 include fish, beef liver and other organ meats, potatoes and other starchy vegetables, and non-citrus fruits [76]. On the other hand, Massey et al. [77] have reported that in individuals with no history of kidney stones, mild to moderate vitamin B6 depletion did not increase urinary oxalate. Rao and Choudhary [78] reported that 10 mg pyridoxine supplementation during 60 days, resulted in a significant decrease in mean 24-h urinary oxalate levels in SF with hyperoxaluria. Curhan et al. [79,80] have found that a high intake of vitamin B6 was inversely associated with the risk of stone formation in women, but not in men. Nevertheless, in a more recent reanalysis of the same cohorts by Ferraro et al. [74] no association between vitamin B6 intake and kidney stones has been disclosed.

In summary, although patients undergoing a BS are under risk of several micronutrient deficiencies [81], there has been no study to date performed among bariatric patients aimed to address the effects of vitamin B6 on preventing kidney stones and the latter is not part of the recommended supplement doses of vitamins after BS as yet.

### 2.4. Vitamin C

Another oxalate-related metabolic pathway is derived from vitamin C (ascorbic acid, ascorbate), an essential micronutrient which humans cannot synthesize due to the lack of the last enzyme in the biosynthetic pathway. The current RDA for vitamin C is 90 mg/day for men and 75 mg/day for women [75]. Although this recommendation can be achieved with a diet rich in fruits and vegetables, ascorbic acid supplements have been widely used for many purposes. Epidemiological data revealed that vitamin C supplements (> 1000 mg/day) were associated with a 16% increase in incidence of kidney stones and increases in oxaluria among men [82]. A metabolic study conducted by our group in adult calcium stone-forming patients [83] has shown a significant increase of 61% and 41% in mean urinary oxalate after taking 1 and 2 g of vitamin C, respectively, while other investigators reported increases of 33% among SF taking 2 g/day [84]. Conversely, Massey et al. [85] found no oxaluric response to vitamin C in many individuals.

In summary, given that vitamin C deficiency may occur after BS [81,86], but harmful effects upon oxaluria can exist, a note of caution should be taken when prescribing ascorbic acid to BS patients, especially for those with a previous history of kidney stones, and urinary oxalate levels must be monitored.

### 2.5. Citrate and Potassium

Hypocitraturia is a common but not a uniform urinary disturbance found after RYGB, ranging from 34% to 63% of patients when present [87,88,89]. The reasons for hypocitraturia have not been fully elucidated since underlying metabolic acidosis, excessive salt and/or animal protein intake have not been observed in most studies [14,87]. Moreover, considering the weight loss after the procedure, low urinary pH induced by obesity should rather be restored to higher values [90].

Dietary management of hypocitraturia can be accomplished by increasing the alkali content of the diet providing higher amounts of vegetables and fruits [91], especially the citric ones such as orange, lemon or lime [92,93,94,95]. Non-citrus alkaline fruits, rich in both citrate and malate also lead to increases in urinary citrate [93], although none of the above have been tested in BS patients. Favorable changes in citrate excretion and urinary pH can be achieved by the intake of mineral water and alkaline beverages depending on their bicarbonate content as well [96]. Finally, a higher intake of potassium provided by fruits and vegetables may also be of help due to the alkaline load [97]. According to Leeman et al. [98], potassium deficiency is very frequent after RYGB procedures, and a recent study showed a decrease of 45.6% on fiber intake after RYGB, reflecting the smaller amount of plant-based food consumption (fruits, vegetables, cereals and legumes), which corresponds to the main source of potassium [99]. Albeit there is no specific recommendation on potassium intake after BS, the RDA for adults is of 120 mEq/day [100]. Pharmacotherapy with Potassium Citrate is another way to control hypocitraturia while raising urinary potassium and pH, in doses of 60 mEq/day or more. Liquid forms seem to be better than pills, for their better absorption than slow-release pills because of the fast gastrointestinal transit [70].

In summary, aiming to increase both potassium and citrate intakes, recommendations after BS should consist of at least 2–3 servings/day of vegetables and fruits (specially the citric ones), based on a nutritional pyramid for this population [101].

### 2.6. Probiotics

Overall, an intense modification in gut microbiota composition regarding phylum, genera and species has been observed after BS [102,103]. According to a recent systematic review, the most noticeable alteration is the overall decrease in the relative abundance of Bifidobacterium and Lactobacillus genera [102], also known as lactic acid bacteria. Both genera have some strains that are acknowledged to play a role in oxalate degradation at the intestinal level as “generalist oxalotrophs”, which degrade alternative carbon sources in addition to oxalate [104]. In a previous study by our group [105] conducted in 14 stone-forming patients, we have tested the effects of a mixture containing *Lactobacillus casei* and *Bifidobacterium breve* upon urinary oxalate reduction after a high-oxalate diet. The effects were extremely variable as we observed different grades of reduction in half of the patients, which in turn seemed to be highly dependent on the concomitant increased response to the dietary oxalate load [106]. A randomized double-blind placebo-controlled trial performed by Goldfarb et al. [107] in idiopathic hyperoxaluric patients did not observe a reduction in urinary oxalate with lactic acid bacteria. A more recent study by Siener et al. [108] also could not find significantly differences on oxaluria after 6 weeks of lactic acid bacteria use in 20 healthy subjects under a high-oxalate diet. The supplementation of probiotics seems to reduce gastrointestinal symptoms in the post-surgery period, favor the increase of vitamin B12 synthesis and potentiate weight loss [109]. Finally, fecal microbiota transplantation (FMT), a procedure in which stool from a healthy donor is placed into another patient’s intestine, has been investigated for the treatment of several diseases, including obesity, with promising results [110]. Although an experimental study showed that the colonization of germ-free mice with stools from RYGB and vertical banded gastroplasty patients induced changes in the microbiota promoting reduced fat deposition in these recipient mice [111], a further clinical investigation could not provide beneficial results with FMT from BS patients [112]. Significantly, it is important to emphasize that there has been no study investigating the effects of FMT in BS patients as an attempt to modify their microbiota or to provide an enriched oxalate-degrading bacteria milieu.

In summary, although the trials with probiotics designed to degrade oxalate in different clinical settings remain still under debate [22,113] and there is no specific data for bariatric patients, the individualized prescription can be considered as an alternative and adjuvant approach that warrants further investigation in terms of dosing, type and timing of administration.

### 2.7. Protein and Sodium

After BS, animal protein consumption is sometimes compromised because of reduced gastric capacity and aversion to certain foods [114]. Golzarand et al. [99] have shown that among patients who underwent RYGB or Sleeve gastrectomy, the reduction of protein intake was around 54% and 65%, respectively, leading to the loss of fat-free mass rather than the desired loss of fat mass. According to the American Society for Nutrition the total protein intake recommendation should be individualized and guided by a registered dietitian, reaching a minimal daily intake of 60 g, with an adequate intake up to 1.5 g/kg/day, considering the ideal body weight (IBW) [115]. A recent review has shown that in most of the studies, patients who had undergone the BS could not reach the amount of protein proposed, suggesting that protein supplementation or diet enrichment could help to achieve the goal of benefiting the patient [114]. On the other hand, a very high protein content in the diet could exert a negative impact on bariatric patients with respect to the risk of kidney stones [97]. Therefore, to maximize the benefits of protein intake adjustments after BS, it is important to consider the patient nutritional needs and monitor their urinary stone risk factors. 

Sodium restriction has an important role in reducing the risk of stone formation. Although dietary sodium limitation has not been evaluated specifically in patients who have undergone BS, there is evidence that a high-sodium diet can increase urinary calcium excretion, decrease urinary citrate excretion and promote the precipitation of calcium salts in the urine, besides worsening bone disease [116,117,118].

In summary, minimal daily intake of 60 g of protein must be achieved by BS patients, and aiming to prevent stone formation an adequate intake of 0.8–1.0 g/Kg IBW/day should be considered. As there is a lack of a particular dietary guidance regarding sodium intake by BS patients, the recommendation for the general population of 2 g/day (5 g of NaCl), according to the World Health Organization [119], might be suggested. Besides salt, the intake of high-sodium food items such as processed and ultra-processed foods should be limited not only to prevent stones but also to help in controlling weight loss after BS [120].

### 2.8. Fluids

It is well established that most BS patients consume a lower amount of fluids due to the small gastric pouch [11,14,88]. In our outpatient unit we have detected a significantly lower urinary volume reflecting such reduced fluid intake even many years following BS [14]. In order to prevent recurrence, SF have been advised to have an appropriated fluid intake, specially water, aimed to achieve a urinary volume of approximately 2.5 L/day or 30 mL/Kg/day [121]. However, there is no specific guidance for BS patients to prevent stone formation, although they are encouraged to drink about 1.8 L/day even when fluid intake is difficult soon after surgery [101]. Moreover, they should avoid high-calorie beverages, such as soft drinks, sport drinks and processed juices, replacing them by water and natural fruit juices. In order to prevent the discomfort or symptoms of Dumping Syndrome, patients can limit fluids to 4 oz (1/2 cup) during mealtimes, and drink liquids around 30 min before eating or 1 h after eating instead.

In summary, assuming the increased risk of kidney stones when the urinary volume is low, BS patients should be recommended to have a fluid intake of around 2.5 L/day. 

### 2.9. Fat

Malabsorptive procedures, such as RYGB, result in fat-malabsorption, contributing to weight loss due to caloric deficit [122]. Besides, it is well established that fat malabsorption is strongly associated with enteric hyperoxaluria and fat-soluble vitamins deficiencies [16]. In a follow up of four years after RYGB, Slater et al. [123] have found an incidence of vitamin A deficiency of 69%, vitamin K deficiency of 68%, and vitamin D deficiency of 63% with altered calcium metabolism. More recently, a ten-year follow-up of 73 patients who underwent biliopancreatic diversion with duodenal switch have also shown a high prevalence of severe deficiency of fat-soluble vitamins [124]. Another complaint of bariatric patients is diarrhea, which can be worsened by the intake of fat [125]. As previously shown in experimental studies conducted by our group [15,31], fat malabsorption may act synergistically with high oxalate intake to produce elevations in urinary oxalate excretion. Recommendations for fat intake after bariatric surgery are similar to those for the general population [115]. Although the range of fat intake calculated from the total energy consumed suggested by the Institute of Medicine is 25%–30% [126], observational studies have shown that fat intake after Sleeve gastrectomy and RYGB surgeries, varies between 37% and 42% [14,99,127]. Recommendations for fat intake after BS are similar to those for the general population [115] and should be implemented aiming to limit fat malabsorption and the above-mentioned effects.

In summary, fat intake recommendation for BS patients should be around 25%–30% of total caloric intake. 

In conclusion, specific guidelines to prevent nephrolithiasis after bariatric surgeries are still lacking. Therefore, the present review, based on the available literature, suggests the main nutritional recommendations for BS patients to reduce the risk of stone formation or recurrence for those who already had stones before the surgery (Figure 2).

## Figures and Tables

**Figure 1 nutrients-12-01442-f001:**
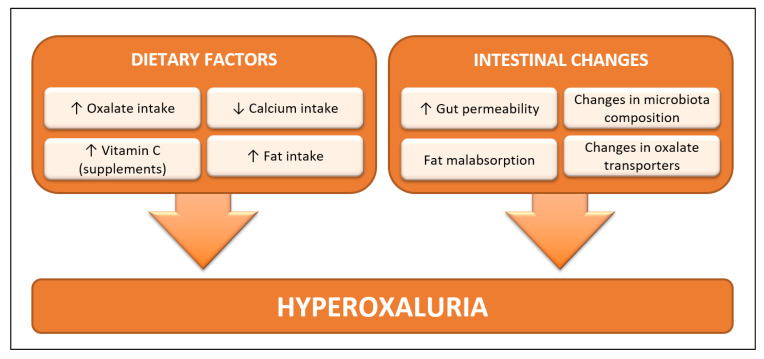
Hypothetical underlying mechanisms for hyperoxaluria after bariatric surgery (BS). A diet rich in oxalate or poor in calcium content decreases the amount of poorly soluble, nonabsorbable calcium oxalate (CaOx) complexes in the intestinal lumen leading to a higher amount of free oxalate for absorption. Vitamin C (ascorbic acid) supplements are metabolized to oxalate contributing to hyperoxaluria. BS predisposes to the development of fat malabsorption, which in the presence of a high dietary fat intake further enhances free oxalate absorption due to the sequestration of calcium by fat. The increased intestinal exposure to complexes of unconjugated bile salts and fatty acids could affect microbiota composition and also decrease the colonization by *Oxalobacter formigenes* and other oxalate-degrading bacteria. Increased gut permeability induced by excessive unconjugated bile salts and other factors may occur. Finally, changes in intestinal oxalate transporters could lead to increased net intestinal oxalate absorption.

**Figure 2 nutrients-12-01442-f002:**
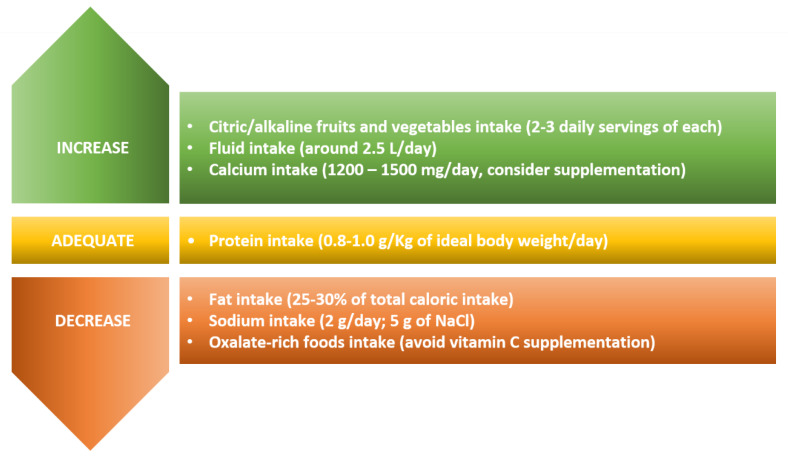
Dietary recommendations for BS patients to prevent the risk of stone formation and to reduce recurrence for those who already had stones before the surgery.

**Table 1 nutrients-12-01442-t001:** Oxalate content in foods (mg/100g).

Food	Description	Oxalate Content (mg/100g)	References
Spinach	Cooked	755–957	[38,50,54]
Spinach	Raw	656–900	[38,50,54]
Rhubarb	Raw	541	[38]
Beet	Roots	76	[38]
Okra	Cooked	45–70	[38,40]
Turnip	Raw	30	[38]
Oca	Cooked	373	[55]
Potato	Baked	24–97	[38,40]
	Chips	75	[38]
	French fries	20–51	[38,40]
Sweet potato	Baked	0.2–86.9	[38,40]
Legumes	Navy Beans	56–76	[38,56]
	Black Beans	71	[56]
	Fava Beans	20	[38]
	Red Kidney Beans	13–26	[38,56]
	Pinto Beans	25–29	[56]
	Soybeans	7.0–57	[38,56]
	Lentils	8.0–39	[38,56]

Star fruit	Raw	80–730	[42,57]
Raspberry	Raw	48	[38]
Orange	Raw	29	[38]
Avocado	Raw	19	[38]

Nuts	Almonds	435–491	[38,56]
	Cashews	175–263	[38,56]
	Walnuts	77–111	[38,56]
	Peanuts	96–148	[38,56]
	Peanut Butter	65	[38]
	Pistachios	46–51	[38,56]
	Pecans	12–66	[38,50,56]
	Sunflower seeds	12	[38]
	Macadamia nuts	40–43	[56]
Bran	Rice bran	281	[38]
	Oat bran	10	[38]
	Wheat bran	34	[38]
	Whole wheat flour	29–67	[38,56]
	White flour	17–41	[38,56]

Chocolate *	Milk chocolate bar ^#^	18–140	[38,56]
	Dark Chocolate bar ^#^	155–485	[58]
	Cocoa powder ^#^	84–783	[38,58]
Coffee *	Filtered	1.0	[38]
	Decaffeinated, filtered	2.0	[38]
Tea *	Black, Brewed	4.0–16	[38,59]
	Green, Brewed	0.3–2.3	[59]

* Variable according to the brand; # variable depending on the amount of cocoa.

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
