# Peer review of "Dietary Recommendations for Bariatric Patients to Prevent Kidney Stone Formation"

_nutrients, 2020, doi:10.3390/nu12051442_

Round 1

Reviewer 1 Report

A well-written manuscript analysing the dietary considerations to prevent kidney stone formation in patients after bariatric surgery.

Some minor comments are below:

1) In order for the reader to better follow the text, please differentiate the nutritional recommendation at the end of each paragraph for every nutrient mentioned.

2) In Figure 2, please be more specific about the recommendation for fruits and vegetables intake,protein, fat and sodium intake. Is there a specific threshold for the above?

Author Response

We thank the Editor and the reviewers for their careful evaluation and the important issues raised about our review article, which helped us to further improve the present manuscript. In its present revised form, the references have been renumbered due to the addition of 4 new references, in response to suggestion by reviewers. The modifications have been highlighted in yellow.

Responses to Reviewer 1 Comments:

Response 1: The reviewer was right. In the present revised form of the manuscript, the nutritional recommendation has been differentiated at the end of each paragraph (lines 142, 189,207,223, 248, 276, 299, 318 and 337). A new reference (Nr.101: “Nutritional pyramid for post-gastric bypass patients”) was added to better support the recommendations for Citrate and Potassium, at line 250.

Response 2: As suggested by the reviewer, the figure 2 was modified, and a specific threshold was added whenever possible.

Reviewer 2 Report

This review is very well-written and articulated. It encompasses a thorough revision of the literature dealing with the original works, most of them showing the factors and main diet elements responsible for stone formation in patients after bariatric surgery. Two figures (one dealing with the hypothetical mechanisms of hyperoxaluria and other proposing dietary recommendations) and a comprehensive table (oxalate content in foods) illustrate the text, which entails 123 references.

This review deals with an area of interest (‘obesity’ management after BS, diet recommendations and evidence) that is of upmost interest for many researchers and clinicians, so in my opinion it could facilitate the apprehension of the great amount of information that is emerging from both in vivo and clinical/epidemiological studies reviewing the hyperoxaluria factor after BS.

I only have minor comments to the review:

1.- Hyperoxaluria can be primary (associated to mutations in AGXT, Serine—pyruvate aminotransferase, or GRHPR, Glyoxylate reductase/hydroxypyruvate reductase) or secondary to another disease process or intervention, as it is focused in this review (i.e. BS). I would expect to have any comment about that within the text, maybe in the Introduction section.

2.- Gut microbiota and probiotics (mainly, lactic acid bacteria) are commented in the review. I do not know if there is any work or study dealing with fecal transplantation in BS patients. It could be of interest.

3.- At some point (line 122), authors use ‘SF’ as an abbreviation. I suppose they refer to ‘Stone Formation’. Please, indicate at first appearance.

4.- Since BS entails restrictive techniques, malabsorptive techniques and combination (i.e. RYGP), I am not sure if the authors could specify more the type of technique used throughout the text instead of using ‘BS’ as a common term.

5.- I think this sentence (line 298) ‘Another complaint of bariatric patients is diarrhea, which can be worsened the intake of protein.’ should be changed by ‘Another complaint of bariatric patients is diarrhea, which can worsen the intake of protein.’

I do not have any further comments to the manuscript. I hope my review finds all the authors in their best condition despite this epidemic outbreak.

Author Response

We thank the Editor and the reviewers for their careful evaluation and the important issues raised about our review article, which helped us to further improve the present manuscript. In its present revised form, the references have been renumbered due to the addition of 4 new references, in response to suggestion by reviewers. The modifications have been highlighted in yellow.

Responses to Reviewer 2 Comments:

Response 1: The reviewer has raised an important point. In the present revised form of the manuscript, a new sentence has been introduced in line 42.

Response 2: The reviewer has raised a valuable suggestion. Although there are few studies about Fecal Microbiota Transplantation (FMT) among patients who had undergone bariatric procedures, we inserted a new paragraph mentioning data available from FMT in the review (line 268) and added the references  110, 111 and 112, to support them.

Response 3: As suggested, the correction of such error was made (now at line 128, in the present form of the manuscript).

Response 4: The reviewer is right. In the present revised form of the manuscript, we have distinguished between both more properly throughout the whole text.

Response 5: We apologize for this mistake and the correction of the missing word was made: “Another complaint of bariatric patients is diarrhea, which can be worsened “by” the intake of fat” (line 328).

Reviewer 3 Report

I have carefully reviewed the manuscript nutrients-801523 with Dr. Ormanji as the first author. The title is " Dietary recommendations for bariatric patients to prevent kidney stone formation”. Bariatric surgery is known to increase risk for nephrolithiasis due to hyperoxaluria. The authors have reviewed some metabolic changes leaded hyperoxaluria found after bariatric surgery and proposed dietary recommendations to prevent the risk of urinary stone formation. Recent recommendations in terms of dietary nutrition to prevent nephrolithiasis were not found. The review manuscript would provide a certain contribution to decrease the risk of urinary stone formation.

Author Response

We thank the Editor and the reviewers for their careful evaluation and the important issues raised about our review article, which helped us to further improve the present manuscript. In its present revised form, the references have been renumbered due to the addition of 4 new references, in response to suggestion by reviewers. The modifications have been highlighted in yellow.

Response to Reviewer 3:

We thank the reviewer for raising such an important point. Although bariatric patients present an increased risk of nephrolithiasis, there is not a specific topic with recommendations to prevent kidney stones for bariatric patients in the general guidelines for nephrolithiasis patients. guidelines. We hope this review may indeed help to shed light into the subject of dietary recommendations for kidney stone prevention for the bariatric population in a more specific and tailored way.